# Modeling of the Natural Objects' Temperature Field Distribution Using a Supercomputer

**Alexander Vitalievich Martirosyan *** and **Yury Valeryevich Ilyushin**

System Analysis and Control Department, Saint Petersburg Mining University, 199106 Saint Petersburg, Russia
* Correspondence: martirosyan320@gmail.com

**Abstract:** There are some regions where the unique type of mineral water mining is compounded by the complex structure conditions of the field. Due to the high emergency risks, the automation of this type of mineral water fields' exploitation process is a necessity. Kislovodskoe mineral water field was chosen as the experimental object, because it has a number of features that make it barely possible to use the conventional methods of mineral water extraction. In the process of extraction, the random vector impacts the mineral water pumping systems. This is mainly due to the complex hydrogeological structure. For the experiment, the laboratory installation and mathematical model were presented by the temperature field changing, due to the similitude of the mathematical apparatus and the hydrodynamic processes behavior,. The main object of this article is the analysis of the reservoir's temperature field behavior using a hybrid supercomputer, and the differences between the supercomputer and a conventional personal computer modeling quality and implementation costs are also given..

**Keywords:** system analysis; computer modeling; observation; mineral water field; hybrid supercomputer





## 1. Introduction

An increase in the number of mineral water bottling enterprises and sanatorium-resort typed organizations has had a significant impact on the mineral water intake growth [1]. Irrational water withdrawal leads to a groundwater quality deterioration, chemical composition and temperature changes [2], and the growth of a depression funnel, which together can lead to the collapse of the reservoir roof and the disappearance of many unique sources [3,4]. This applies to all mineral water fields located in the region of the Caucasian Mineral Waters [5]. Due to the mentioned above, there is a potential danger of the mineral water fields' degradation [6,7].

Therefore, an important task is to create predictive models for the development of hydrodynamic processes with a change in the volume of water intake in various parts of the field [8,9].

## 2. Problem Statement

Compilation of hydrodynamic models at the stage of synthesis of closed distributed control systems is a particularly difficult task [10,11], mainly due to the fact that modeling of spatially invariant situations expressed in terms of spatial delta functions for multidimensional objects is not subject to analytical solutions [12]. When modeling such types of systems, it is necessary to create simplified (truncated) customary mathematical models; however, such models differ significantly from real physical processes [13,14]. In order to reduce the error of this system, the finite-dimensional mathematical model is taken to be excessively large, which leads to a large number of calculations, but brings the mathematical model as close as possible to the physical process [15,16]. There are mathematical models which are performed on personal computers using various programming languages; however, such mathematical operations require large expenditures for

modeling [17,18]. Modern technologies make it possible to combine groups of personal computers into complex computing systems that can significantly increase the speed of calculations and, accordingly, the control module's response time of the nature or intensity of input action changes [19,20]. Until recently, such computer systems were the main means of technical processes' mathematical modeling, including automatic control systems modeling [21,22]. High-performance computing systems based on NVidia graphics cards have appeared on the computer systems market. Mathematical operations on such calculators are performed quite quickly, mainly due to the fact that these calculators are adapted for parallel data processing [23,24]. The main goal of this article is to observe the behavior of the temperature field formed as a result of the occurrence of different temperatures of narzan at different depths on a hybrid supercomputer based on a 96-core processor NVidia GF 104 (nVIDIA, State of California, USA). This task is relevant because of the urgent need to take into account the amount of narzan at different depths of mineral water [25,26].

## 3. Research Methodology

With the development of information technologies, more accurate methods for numerical modeling of both stationary and non-stationary geofiltration processes in multilayer systems appear, allowing you to set the initial and boundary conditions of the I, II, and III kind, as well as such characteristics of the filtration flow as the elastic flow regime in weakly permeable clayey interlayers, heterogeneity of the filtering fluid in the reservoirs, etc. [27,28]. Such applications include Topaz, PLAST, Simulation CFD, MIF—3D, ModTech, etc. Such information systems allow you to build fairly complex models of hydrogeological processes, but they are available only as executable applications, so it is impossible to create a regulator for these models and simulate a control action [29,30].

Among the studies of recent years, the work of A.V. Malkov and I.M. Pershin is of particular importance [7]. In this study, the tasks of improving the theoretical foundations of managing natural geological objects under conditions of intense technogenic impact, increasing the accuracy and efficiency of managing the processes of field operation, and substantiating development conditions with minimal environmental impact are solved. The book discusses the general principles of building hydrodynamic models, methods for determining the hydrodynamic parameters of aquifers, calculating optimal operating modes, the fundamentals of the theory of analysis and synthesis of systems with distributed parameters, and provides solutions to practical problems using the example of the Georgievskoye and Kuyulusskoye fields.

As a result of the hydrodynamic processes' modeling problem exploration degree analysis, it can be concluded that, despite the relevance of the topic, modeling problems are solved only partially for individual areas. Modeling a large region, for which it is required to set heterogeneous initial and boundary conditions, seems to be a large-scale problem that requires large computational resources [31,32].

Therefore, the next stage in the study of modeling problems is the construction of a mathematical and computer model of a block structure, and optimization of software that implements this model [33]. This will make it possible to build regulators to control the process of water intake of a large hydrogeological object.

The main problem in the development of mathematical and computer models is that it is necessary to find a compromise: on the one hand, it is necessary to take into account as many factors as possible that affect the physical process, since it is required to describe the real picture as fully as possible so that the model is as adequate as possible, corresponding to the original object. On the other hand, the complication of the mathematical, and, as a consequence, the computer model, makes it impossible to implement it by means of a computer, since such a model requires too much computer resources, which significantly affects the cost of installing, maintaining, and operating equipment [34,35]. In this regard, the second task of the study was to create a parallel algorithm which allows performing calculations more efficiently for a fairly complex model.

For this purpose, it is necessary to understand how the approach to creating a parallel program on a computing cluster differs from programming on a hybrid supercomputer. The core of the NVidia GF 104 processor consists of a common data exchange bus, which includes a shared 2 GB cache, directly connected to 96 arithmetic logic units.

Thus, when modeling a technological process, it is necessary to divide the mathematical model into functional components, which, in turn, using the extension of the C programming language for hybrid supercomputers, will be redirected to various arithmetic logic units. Let us move on to solving the mathematical Model (1) and adapting it for computing on a hybrid supercomputer based on the NVidia GF 104 processor.

When calculating such fields, second-order partial differential equations are also used. This equation looks like this:

$$\frac{\partial T(x,y,z,\tau)}{\partial t} = a\left(\frac{\partial^2 T(x,y,z,\tau)}{\partial x^2} + \frac{\partial^2 T(x,y,z,\tau)}{\partial y^2} + \frac{\partial^2 T(x,y,z,\tau)}{\partial z^2}\right) + Q_{giv} \qquad (1)$$

where $T$ is temperature; $t$—time; $a$—coefficient of thermal diffusivity; $Q_{giv}$ is the power of internal heat sources.

Having calculated the temperature field based on the parallel synthesis algorithm, a discrete analogue of the pathetic model was obtained:

The discrete analogue of the mathematical model of the temperature field is presented as follows:

$$T(x,y,z,\tau) = a \cdot \Delta t \left( \begin{array}{l} \frac{T(x-1,y,z,t)-2T(x,y,z,t)+T(x+1,y,z,t)}{\Delta x^2} + \\ + \frac{T(x,y-1,z,t)-2T(x,y,z,t)+T(x,y+1,z,t)}{\Delta y^2} + \\ + \frac{T(x,y,z-1,t)-2T(x,y,z,t)+T(x,y,z+1,t)}{\Delta z^2} \end{array} \right) + Q_{giv} \qquad (2)$$

Since a feature of the Kislovodsk narzan field is the presence of different types of narzan at one geographical point (sulphate, dolomite, general), it is necessary to calculate the values of temperature fields for each layer of narzan. To this end, discrete analogues for three types of narzan located one above the other were obtained [36].

Thus, the layers will take the form shown in Figure 1 [2]. The figure shows that the model of Kislovodsk mineral water field formations indicates: 1—common narzan; 2—dolomite; 3—sulfate.

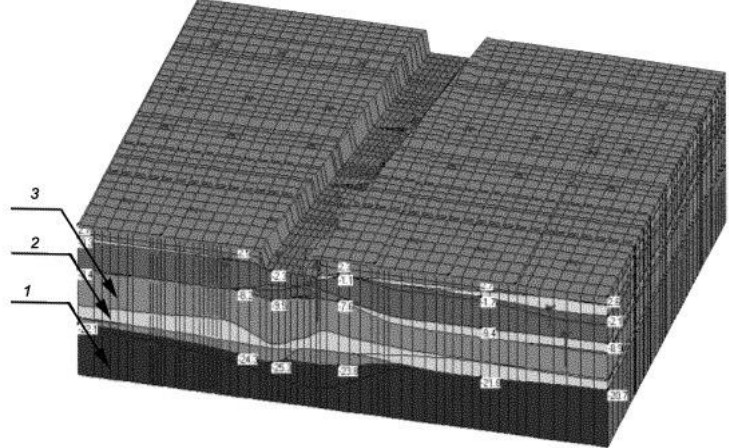

**Figure 1.** Location of aquifers [2].

Let us calculate the temperature field on a hybrid supercomputer (Algorithm 1). To calculate the temperature field, the following variables had been set [8,9]:

---

**Algorithm 1.** Fields temperature calculation.

---

```
int l=10;
int timer, timer_1;
int r, fi, x;
double dT[I][J][K][M];
double ****dev_dT;
double a, lam1=0.00038, lam2=0.00003026, N;
```
Next, you need to allocate memory on the GPU, into which the data for the calculation will later be copied.
```
HANDLE_ERROR( cudaMalloc( (void**)&dev_dT, I*J*K*M *sizeof( float ) ) );
```
One of the main difficulties of working with memory is the allocation of a memory array, which will be copied to the CPU for subsequent display. The program code that copies the result from the GPU to the CPU is as follows:
```
HANDLE_ERROR( cudaMemcpy( dT, dev_dT, I*J*K*M *sizeof( float ), cudaMemcpyDeviceToHost ) );
        for (r=1; r<=8; r++)
            for (fi=1; fi<=30; fi++)
                for (x=1; x<=1199; x++)
                {
                 printf ("%d %d %d %d \n", r, fi, x, dT[r][fi][x]);
                }
        HANDLE_ERROR( cudaFree( dev_dT ) );
        return 0;
}
```
Code generated for functions such as the kernel runs on the GPU.
```
__global__ void Cuda_proccedure(double ****dT)
{
        for (r=1; r<=8; r++)
            for (fi=1; fi<=30; fi++)
                    for (x=1; x<=1199; x++)
                {
                    T[r][fi][x][timer]=0;
                    dT[r][fi][x][timer]=0;
                }
        for (r=1; r<=I-1; r++)
            for (fi=1; fi<=30; fi++)
                for (x=1; x<=1199; x++)
                {
                    if ((r<5)||(r>8)) {a=22.16;}
                    else a=0.3769;

dT[r][fi][x][timer]=((T[r-1][fi][x][timer]-2*T[r][fi][x][timer]+T[r+1][fi][x][timer])/r*r+(T[r][fi-1][x][timer]-
2*T[r][fi][x][timer]+T[r][fi+1][x][timer])/fi*fi+(T[r][fi][x-1][timer]-
2*T[r][fi][x][timer]+T[r][fi+1][x+1][timer])/x*x);
                }
                        for (r=0; r<=9; r++)
        for (fi=0; fi<=30; fi++)
                {
                T[r][fi][0][timer]=10;
                T[r][fi][30][timer]=T[r][fi][30][timer];
                }
        for (fi=0; fi<=359; fi++)
            for (x=1; x<=1199; x++)
                {
                T[0][fi][x][timer]=T[1][fi][x][timer];
                T[5][fi][x][timer]=(lam1*T[6][fi][x][timer]-lam2*T[4][fi][x][timer])/(lam1+lam2);
                T[9][fi][x][timer]=T[4][fi][x][timer];
                T[8][fi][x][timer]=(lam1*T[7][fi][x][timer]-lam2*T[9][fi][x][timer])/(lam1+lam2);
                }
****************************************************
}
```

---

## 4. Results

According to the purpose of the study it is necessary to identify and calculate the linear connection between narzan water temperature fields' distribution. The object's parameters $l = 10$, $k = 10$, $d = 9$, $T_{giv} = 0.3$, $a^2 = 0.01$ are given as an initial datum for the modeling experiment. The obtained results with the different heating points are given in Table 1.

**Table 1.** The results of the shallow lying aquifer modeling.

| d = 9 | d = 8 | d = 7 | d = 6 | d = 5 |
|---|---|---|---|---|
| tp [1,690] = 0.19 | tp [1,690] = 0.19 | tp [1,690] = 0.18 | tp [1,690] = 0.18 | tp [1,690] = 0.48 |
| tp [2,690] = 0.37 | tp [2,690] = 0.36 | tp [2,690] = 0.31 | tp [2,690] = 0.32 | tp [2,690] = 0.39 |
| tp [3,690] = 0.49 | tp [3,690] = 0.47 | tp [3,690] = 0.43 | tp [3,690] = 0.37 | tp [3,690] = 0.39 |
| tp [4,690] = 0.56 | tp [4,690] = 0.51 | tp [4,690] = 0.43 | tp [4,690] = 0.32 | tp [4,690] = 0.38 |
| tp [5,690] = 0.56 | tp [5,690] = 0.47 | tp [5,690] = 0.34 | tp [5,690] = 0.18 | tp [5,690] = 0.45 |
| tp [6,690] = 0.49 | tp [6,690] = 0.36 | tp [6,690] = 0.19 | tp [6,690] = 0.26 | |
| tp [7,690] = 0.37 | tp [7,690] = 0.19 | tp [7,690] = 0.42 | | |
| tp [8,690] = 0.19 | tp [8,690] = 0.78 | | | |
| tp [9,690] = 0.14 | | | | |
| **d = 14** | **d = 13** | **d = 12** | **d = 11** | **d = 10** |
| tp [1,690] = 0.20 | tp [1,690] = 0.20 | tp [1,690] = 0.19 | tp [1,690] = 0.19 | tp [1,690] = 0.19 |
| tp [2,690] = 0.39 | tp [2,690] = 0.38 | tp [2,690] = 0.38 | tp [2,690] = 0.38 | tp [2,690] = 0.37 |
| tp [3,690] = 0.56 | tp [3,690] = 0.55 | tp [3,690] = 0.54 | tp [3,690] = 0.53 | tp [3,690] = 0.51 |
| tp [4,690] = 0.70 | tp [4,690] = 0.68 | tp [4,690] = 0.66 | tp [4,690] = 0.64 | tp [4,690] = 0.60 |
| tp [5,690] = 0.80 | tp [5,690] = 0.77 | tp [5,690] = 0.74 | tp [5,690] = 0.69 | tp [5,690] = 0.63 |
| tp [6,690] = 0.87 | tp [6,690] = 0.82 | tp [6,690] = 0.76 | tp [6,690] = 0.69 | tp [6,690] = 0.60 |
| tp [7,690] = 0.89 | tp [7,690] = 0.82 | tp [7,690] = 0.74 | tp [7,690] = 0.64 | tp [7,690] = 0.51 |
| tp [8,690] = 0.87 | tp [8,690] = 0.77 | tp [8,690] = 0.66 | tp [8,690] = 0.53 | tp [8,690] = 0.37 |
| tp [9,690] = 0.80 | tp [9,690] = 0.68 | tp [9,690] = 0.54 | tp [9,690] = 0.38 | tp [9,690] = 0.19 |
| tp [10,690] = 0.70 | tp [10,690] = 0.55 | tp [10,690] = 0.38 | tp [10,690] = 0.19 | tp [10,690] = 0.50 |
| tp [11,690] = 0.56 | tp [11,690] = 0.38 | tp [11,690] = 0.19 | tp [11,690] = 0.85 | |
| tp [12,690] = 0.39 | tp [12,690] = 0.20 | tp [12,690] = 0.21 | | |
| tp [13,690] = 0.20 | tp [13,690] = 0.56 | | | |
| tp [14,690] = 0.92 | | | | |

The investigation of the object with an average occurrence of the mineral layer: $l = 10$, $k = 10$, $d = 9$, $T_{giv} = 0.3$, $a^2 = 0.1$ is given in Table 2.

**Table 2.** The results of the middle-deep lying aquifer modeling.

| d = 9 | d = 8 | d = 7 | d = 6 | d = 5 |
|---|---|---|---|---|
| tp [1,690] = 2.03 | tp [1,690] = 2.01 | tp [1,690] = 1.99 | tp [1,690] = 1.95 | tp [1,690] = 1.89 |
| tp [2,690] = 3.82 | tp [2,690] = 3.73 | tp [2,690] = 3.59 | tp [2,690] = 3.39 | tp [2,690] = 3.07 |
| tp [3,690] = 5.15 | tp [3,690] = 4.87 | tp [3,690] = 4.48 | tp [3,690] = 3.91 | tp [3,690] = 3.07 |
| tp [4,690] = 5.86 | tp [4,690] = 5.27 | tp [4,690] = 4.48 | tp [4,690] = 3.39 | tp [4,690] = 1.89 |
| tp [5,690] = 5.86 | tp [5,690] = 4.87 | tp [5,690] = 3.59 | tp [5,690] = 1.95 | tp [5,690] = 1.75 |
| tp [6,690] = 5.15 | tp [6,690] = 3.73 | tp [6,690] = 1.99 | tp [6,690] = 2.12 | |
| tp [7,690] = 3.82 | tp [7,690] = 2.01 | tp [7,690] = 2.49 | | |
| tp [8,690] = 2.03 | tp [8,690] = 2.86 | | | |
| **d = 14** | **d = 13** | **d = 12** | **d = 11** | **d = 10** |
| tp [1,570] = 2.07 | tp [1,570] = 2.06 | tp [1,570] = 2.06 | tp [1,570] = 2.05 | tp [1,570] = 2.04 |
| tp [2,570] = 4.04 | tp [2,570] = 4.01 | tp [2,570] = 3.98 | tp [2,570] = 3.94 | tp [2,570] = 3.89 |
| tp [3,570] = 5.80 | tp [3,570] = 5.73 | tp [3,570] = 5.63 | tp [3,570] = 5.51 | tp [3,570] = 5.36 |
| tp [4,570] = 7.28 | tp [4,570] = 7.11 | tp [4,570] = 6.90 | tp [4,570] = 6.64 | tp [4,570] = 6.30 |
| tp [5,570] = 8.39 | tp [5,570] = 8.08 | tp [5,570] = 7.70 | tp [5,570] = 7.22 | tp [5,570] = 6.62 |
| tp [6,570] = 9.08 | tp [6,570] = 8.58 | tp [6,570] = 7.97 | tp [6,570] = 7.22 | tp [6,570] = 6.30 |
| tp [7,570] = 9.31 | tp [7,570] = 8.58 | tp [7,570] = 7.70 | tp [7,570] = 6.64 | tp [7,570] = 5.36 |
| tp [8,570] = 9.08 | tp [8,570] = 8.08 | tp [8,570] = 6.90 | tp [8,570] = 5.51 | tp [8,570] = 3.89 |
| tp [9,570] = 8.39 | tp [9,570] = 7.11 | tp [9,570] = 5.63 | tp [9,570] = 3.94 | tp [9,570] = 2.04 |
| tp [10,570] = 7.28 | tp [10,570] = 5.73 | tp [10,570] = 3.98 | tp [10,570] = 2.05 | tp [10,570] = 3.59 |
| tp [11,570] = 5.80 | tp [11,570] = 4.01 | tp [11,570] = 2.06 | tp [11,570] = 3.95 | |
| tp [12,570] = 4.04 | tp [12,570] = 2.06 | tp [12,570] = 4.32 | | |
| tp [13,570] = 2.07 | tp [13,570] = 4.68 | | | |
| tp [14,570] = 2.07 | | | | |

Let the initial data been changed to meet the new requirements and set the following values: $l = 10$, $k = 10$, $d = 9$, $T_{giv} = 0.3$, $a^2 = 0.2$. The results are given in Table 3.

**Table 3.** The results of the deep lying aquifer.

| d = 9 | d = 8 | d = 7 | d = 6 | d = 5 |
|---|---|---|---|---|
| tp [1,690] = 0.11 | tp [1,690] = 0.37 | tp [1,690] = 0.33 | tp [1,690] = 0.23 | tp [1,690] = 0.24 |
| tp [2,690] = 0.21 | tp [2,690] = 0.57 | tp [2,690] = 0.60 | tp [2,690] = 0.38 | tp [2,690] = 0.39 |
| tp [3,690] = 0.29 | tp [3,690] = 0.59 | tp [3,690] = 0.73 | tp [3,690] = 0.43 | tp [3,690] = 0.39 |
| tp [4,690] = 0.33 | tp [4,690] = 0.58 | tp [4,690] = 0.73 | tp [4,690] = 0.38 | tp [4,690] = 0.24 |
| tp [5,690] = 0.33 | tp [5,690] = 0.59 | tp [5,690] = 0.60 | tp [5,690] = 0.23 | tp [5,690] = 0.24 |
| tp [6,690] = 0.29 | tp [6,690] = 0.57 | tp [6,690] = 0.33 | tp [6,690] = 0.84 | |
| tp [7,690] = 0.21 | tp [7,690] = 0.37 | tp [7,690] = 0.41 | | |
| tp [8,690] = 0.11 | tp [8,690] = 0.02 | | | |
| tp [9,690] = 0.84 | | | | |

| d = 14 | d = 13 | d = 12 | d = 11 | d = 10 |
|---|---|---|---|---|
| tp [1,690] = 0.19 | tp [1,690] = 0.17 | tp [1,690] = 0.24 | tp [1,690] = 0.09 | tp [1,690] = 0.10 |
| tp [2,690] = 0.38 | tp [2,690] = 0.33 | tp [2,690] = 0.45 | tp [2,690] = 0.19 | tp [2,690] = 0.19 |
| tp [3,690] = 0.52 | tp [3,690] = 0.46 | tp [3,690] = 0.62 | tp [3,690] = 0.26 | tp [3,690] = 0.26 |
| tp [4,690] = 0.64 | tp [4,690] = 0.55 | tp [4,690] = 0.74 | tp [4,690] = 0.32 | tp [4,690] = 0.31 |
| tp [5,690] = 0.71 | tp [5,690] = 0.61 | tp [5,690] = 0.80 | tp [5,690] = 0.34 | tp [5,690] = 0.33 |
| tp [6,690] = 0.75 | tp [6,690] = 0.63 | tp [6,690] = 0.82 | tp [6,690] = 0.34 | tp [6,690] = 0.31 |
| tp [7,690] = 0.76 | tp [7,690] = 0.63 | tp [7,690] = 0.80 | tp [7,690] = 0.32 | tp [7,690] = 0.26 |
| tp [8,690] = 0.75 | tp [8,690] = 0.61 | tp [8,690] = 0.74 | tp [8,690] = 0.26 | tp [8,690] = 0.19 |
| tp [9,690] = 0.71 | tp [9,690] = 0.55 | tp [9,690] = 0.62 | tp [9,690] = 0.19 | tp [9,690] = 0.10 |
| tp [10,690] = 0.64 | tp [10,690] = 0.46 | tp [10,690] = 0.45 | tp [10,690] = 0.09 | tp [10,690] = 0.82 |
| tp [11,690] = 0.52 | tp [11,690] = 0.33 | tp [11,690] = 0.24 | tp [11,690] = 0.93 | |
| tp [12,690] = 0.38 | tp [12,690] = 0.17 | tp [12,690] = 0.69 | | |
| tp [13,690] = 0.19 | tp [13,690] = −0.7 | | | |
| tp [14,690] = 0.61 | | | | |

The results of mathematical modeling on a supercomputer and on a conventional personal computer are shown in Table 4. As can be seen from the table, the values are identical, but the time spent on the calculation is much less.

**Table 4.** Mathematical modeling results.

| Step | Supercomputer Results | Regular PC Results |
|---|---|---|
| 1 | 12.38692 | 12.38692 |
| 2 | 12.38892 | 12.38892 |
| 3 | 13.39692 | 13.39692 |
| . . . . . . | . . . . . . | . . . . . . |
| 100 | 87.19692 | 87.19692 |
| 101 | 89.31292 | 89.31292 |
| 102 | 97.29692 | 97.29692 |

Consequently, mathematical modeling of automatic control systems will significantly increase the performance of systems. To a greater extent, this is necessary for real-time systems. Obtaining the same results can be done in much less time, which is important for faster system reactions. Despite the fact that hydrodynamic processes are slow, when we observe a mineral water field with closely spaced wells it must be assumed not only the necessity of fast system reaction but also the inertness of control action. The control system must simultaneously analyze the current situation and also predict the possible variations in the controlled magnitudes. A personal computer does not have enough capacity for such complex processes support. This goes to show the adequacy of the proposed automation approach with relation to the problems of mineral water fields exploitation area.

## 5. Discussion

The ecological resort area of the Caucasian Mineral Waters (KMV) occupies a special place among the resort regions of Russia due to the wealth, diversity, quantity, and value of mineral waters, landscape and climatic conditions, and therapeutic mud. Recently, the rate of mineral water resources development for an industrial bottling and resort purposes has increased.

An increase in the number of mineral water bottling enterprises and organizations of a sanatorium-resort type has had a significant impact on the growth in the volume of mineral water intake. Irrational water withdrawal leads to a deterioration in the quality of groundwater, a change in their chemical composition, temperature, and the growth of a depression funnel, which can lead to the collapse of the roof of the reservoir and the disappearance of many sources from the face of the Earth. This applies to all waters located in the Caucasian Mineral Waters (CMV) region. In view of the above, there is a potential danger of degradation of the mineral water fields under consideration.

Therefore, an important task is to create predictive models for the development of hydrodynamic processes in the region with a change in the volume of water intake in various parts of the field. The paper proposes a method for controlling the level of the aquifer in production wells that provides a given decrease in the water level. The reliability of the manuscripts' scientific results is confirmed by the correct use of the mathematical apparatus in the description and study of the hydrodynamic processes of the mineral water field, as well as by the consistency of the results of theoretical studies and computer simulation of the obtained control systems.

The scientific novelty of the work lies in the mathematical description of the relationship between the main factors and parameters of the aquifers, in the synthesis of a distributed control system for the parameters of a depression funnel.

## 6. Conclusions

Due to the aggravating condition of the hydromineral base of the considered mineral water field and the whole CMV region, the automation of the exploitation processes becomes a necessity and the only chance to save the unique water types. In the extraction industries of the region there are only few cases of partial automation and control systems used for improving the technological process. This is already a positive step in this area, but controlling the single-unit wells does not give the opportunity of the whole field condition monitoring. The popularity of this water type has created the necessity for high flow rate extraction, which can influence the fluid level of the whole field. It is impossible to neglect the interconnection between the wells of the one aquifer, but the attempts to develop a system of monitoring on the first step (and then a control system) will meet problems with the huge amount of calculations. Due to this fact, the regular computer used will be effective only in single-unit automation, but inapplicable in complex monitoring. That is why, for the big industries which have adequate funds, it becomes much more effective to use supercomputers versus personal ones. The present research was undertaken to prove the possibility and effectiveness of the proposed automation approach.

Using the laboratory installation, which can imitate the structure and conditions of the mineral water field, the experiment was carried out. For the mathematical modeling the real data of Kislovodkoe field were used. The approbation of the developed model shows the high convergence measure of the model and real object. Then, the reaction of the system with the same input actions was equated both on a supercomputer and a regular one.

Based on the results of mathematical modeling of the control object's heat flow patterns, the following conclusions can be drawn.

1.  The increasing of the mineral layer's heating points number both increases the computational complexity and modeling accuracy, which is so important in the case of complex structured object exploitation, because if the field's structure entirety or exploitation process sustainability are disturbed, it causes to the possibility of damaging

2.     the well or the whole aquifer. Assuming this, it can be concluded that the use of a supercomputer for the hydrodynamic modeling processes is justified.

2. When modeling the process, it was seen that the heating process does not proceed uniformly, but depends on the height above sea level. It can also be concluded that it is necessary to develop methods for analyzing and constructing observers in the design of drilling rigs for the extraction of mineral water.

3. Simulation of this process on a hybrid supercomputer led to a real calculation time of 10 min. In turn, a linear algorithm on a conventional computer simulates this process for 6 h. This speed is realized mainly due to the 96-core processor, which operates at a higher total frequency than a conventional personal computer processor. It completes the equations approximately sixty times faster, and its implementation will cost only eight to ten times more than a regular one.

The analysis of the results of mathematical modeling shows that the modeling of automatic control systems and forecasting systems, including robust control, can be significantly expanded through the use of hybrid supercomputers. Their application will allow not only expanding the control actions' number, but also the time of transient responses modeling.

**Author Contributions:** Conceptualization, Y.V.I. and A.V.M.; methodology, A.V.M.; software, Y.V.I.; validation, A.V.M.; formal analysis, Y.V.I.; investigation, A.V.M.; resources, Y.V.I.; data curation, A.V.M.; writing—original draft preparation, A.V.M.; writing—review and editing, Y.V.I. and A.V.M.; visualization, A.V.M.; supervision, Y.V.I.; project administration, A.V.M.; funding acquisition, Y.V.I. All authors have read and agreed to the published version of the manuscript.

**Funding:** This research received no external funding.

**Institutional Review Board Statement:** Not applicable.

**Informed Consent Statement:** Not applicable.

**Conflicts of Interest:** The authors declare no conflict of interest.

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
