# Peer review of "Modeling of the Natural Objects’ Temperature Field Distribution Using a Supercomputer"

_informatics, doi:10.3390/informatics9030062_

Round 1

Reviewer 1 Report

Abstract Is not complete. It needs to give the clear information about the output (numerical interpretation is required).

 Literature review is not adequate to emphasize the problem and statement.

 Also lack of references in the background and problem statement.

 Figure 1: need to replace. Since it is not produced by the writer it should have the references and also the permission

Result section was not properly presented. Needs to put proper explanation to support this output. 

Conclusion must need to improve.

Author Response

Thank you for your feedback.

We tried to correct all the remarks carefully.

1. Abstract Is not complete. It needs to give the clear information about the output (numerical interpretation is required).

Abstract had been improved. Short numerical interpretation of the result was given.

2. Literature review is not adequate to emphasize the problem and statement. Also lack of references in the background and problem statement.

The problem is that there is not a lot of publications in this scientific area. Despite the huge amount of publications about hydrosphere modeling, there are only few among them are related to control systems and technological-process automation. The intercomparison of regular and super-computer using efficiency of automation problem solving was not considered before. Due to above-mentioned, the most of references are closely connected with different parts of research, all the given articles partly describes all kinds of science aspects of submitted paper.

The background and problem statement is best documented in the sources 2, 5, 7.

Using high speed calculating computers 23, 24.

3. Figure 1: need to replace. Since it is not produced by the writer it should have the references and also the permission

The given figure was designed as a part of collaborative work with Malkov, A.V (References, №2) who is the general director of “Narzan Giroresursy” and our coworker. All the experiments had been done with the real data. Anyway, we have added the reference to a figure.

4. Result section was not properly presented. Needs to put proper explanation to support this output. Conclusion must need to improve.

Conclusion of the article was improved. The results are also shown better now.

Reviewer 2 Report

General comments

1.     The subject of the article complies with the A&S of the Journal MDPI Informatics.

2.     Presented research meets a highly topical issue.

3.     The detailed description of obtained results is sufficiently complete.

4.     The manuscript can be recommended to publication in MDPI «Informatics» after the minor revision.

Comments on manuscript

1.     There are a number of lexical mistakes which distorts the meaning of what is written. The manuscript needs to be clearly rewritten in correct English.

2.     The style of the article is closer to experiment results report then to research article. 

3.     Specific technical possibilities of applying the presented results are not clearly presented in the text.

4.     It is not entirely clear how the efficiency of the distributed systems synthesis\implementation is connected with the super-computer using. Why the presented model of super-computer is proposed? Implementation of the super-computer to a regular technological process is a complex task, which needs major changes in data collection and processing process. How profitable is the introduction of proposed technologies? Some definitions or recommendations must be given in the text of manuscript.

5.     The text from lines 227-235 must be revised. Material should be presented in a coherent and consistent manner.

Author Response

Thank you for your feedback.

We tried to correct all the remarks carefully.

All the corrections are in the attached file.

Comments on manuscript

  1. There are a number of lexical mistakes which distorts the meaning of what is written. The manuscript needs to be clearly rewritten in correct English.

The text of the paper was carefully revised.

  1. The style of the article is closer to experiment results report then to research article. 

Parts of the article which shows the methods and results were revised.

  1. Specific technical possibilities of applying the presented results are not clearly presented in the text.

Parts of the article which shows the methods and results were revised.

  1. It is not entirely clear how the efficiency of the distributed systems synthesis\implementation is connected with the super-computer using. Why the presented model of super-computer is proposed? Implementation of the super-computer to a regular technological process is a complex task, which needs major changes in data collection and processing process. How profitable is the introduction of proposed technologies? Some definitions or recommendations must be given in the text of manuscript.

All the mentioned aspects were added to conclusion.

  1. The text from lines 227-235 must be revised. Material should be presented in a coherent and consistent manner.

This part of the article was fully revised.

Reviewer 3 Report

Important comments:

 There is a fair amount of grammar and syntax mistakes, which should be carefully improved.

 The title of the manuscript is not fully correct – given research shown the mathematical model of the real object (mineral water field), where the temperature in aquifers plays the role only in extraction process. While the experimental modeling apparatus is correct, you must mention, that temperature distribution matters only in experiments and doesn’t work so in real object.

 In title you must choose one: analysis of temperature distribution in modeling apparatus or analysis or modeling of the real object, but without temperature distributions.

  You must show the technical characteristics of the both used computers to make the results clearer.

 Implementation of that sort on programs\systems is always very expensive. What are the advantageous of there using?

 Line 54 What does “mixed narzan” meaning? How it can be possible?

 Line 147 The Russian text must be avoided.

 Line 166 What does the “a” means and how this value was determined?

Recommendations:

 It would be better to identify the language of the program and argue its using for the solution of this problem.

 The given program is an experimental prototype or a released product?

Author Response

Thank you for your feedback.

We tried to correct all the remarks carefully.

All the corrections are in the attached file.

Important comments:

1. There is a fair amount of grammar and syntax mistakes, which should be carefully improved.

The text of the paper was carefully revised.

 2. The title of the manuscript is not fully correct – given research shown the mathematical model of the real object (mineral water field), where the temperature in aquifers plays the role only in extraction process. While the experimental modeling apparatus is correct, you must mention, that temperature distribution matters only in experiments and doesn’t work so in real object. In title you must choose one: analysis of temperature distribution in modeling apparatus or analysis or modeling of the real object, but without temperature distributions.

The title of the paper was carefully revised.

 You must show the technical characteristics of the both used computers to make the results clearer.

Configuration of regular PC is as follows: Intel® Core™ i5-12400; Radeon RX 480; RAM 16GB; 200 GB SSD.

The super-computer is a property of university and now my coworkers and me don’t have an access to it due to the summer season. Approximately configuration is NVIDIA GF104 GPU; GeForce GTX 770; RAM 64GB; 1 TB SSD. It would be very hard now to receive the exact information.

This information was not added to a manuscript text because we did not find it useful for a possible audience. Anyway, if somebody will try this experiment, they would do it on the machines with there own configurations. Proposed idea was not planned as a box solution, it’s more about the calculation-centered tools using for the improving of control efficiency. If you think that this information is absolutely necessary, it would be added.

3. Implementation of that sort on programs\systems is always very expensive. What are the advantageous of there using?

All the advantages are now shown in conclusion. In addition to it: in the case where automation not only a production improving instrument, but a necessity, the proposed approach can save a lot of money and time, because it already is the step forward technical solution which exactly would be advantageous.

4. Line 54 What does “mixed narzan” meaning? How it can be possible?

Sorry, that was a mistake. Everything was already corrected.

5. Line 147 The Russian text must be avoided.

All the corrections had been done.

 6. Line 166 What does the “a” means and how this value was determined?

  “a” is coefficient of thermal diffusivity. In heat transfer analysis, thermal diffusivity is the thermal conductivity divided by density and specific heat capacity at constant pressure. It measures the rate of transfer of heat of a material from the hot end to the cold end.

Recommendations:

7. It would be better to identify the language of the program and argue its using for the solution of this problem.

The shown part of a program is on C. It was added to a text. Line 194

8. The given program is an experimental prototype or a released product?

The given program is an experimental prototype, tested only on a laboratory installation. At that moment there are some difficulties with an experiment on the real object, but we are planning to do it towards the end of the year.

Round 2

Reviewer 1 Report

Thanks for update this article.